# Corticosterone and testosterone treatment influence expression of gene pathways linked to meiotic segregation in preovulatory follicles of the domestic hen

**Elizabeth R. Wrobel**[1‡], **Alexandra B. Bentz**[2‡], **W. Walter Lorenz**[3], **Stephen T. Gardner**[4], **Mary T. Mendonça**[4], **Kristen J. Navara**[1] *

1 Department of Poultry Science, The University of Georgia, Athens, GA, United States of America, 2 Department of Biology, Indiana University, Bloomington, IN, United States of America, 3 Institute of Bioinformatics and Georgia Genomics and Bioinformatics Core, The University of Georgia, Athens, GA, United States of America, 4 Department of Biological Sciences, Auburn University, Auburn, AL, United States of America

‡ These authors share first authorship on this work.
* knavara@gmail.com

**Data Availability Statement:** Data are included as supplemental files, and the genomic data have

## Abstract

Decades of work indicate that female birds can control their offspring sex ratios in response to environmental and social cues. In laying hens, hormones administered immediately prior to sex chromosome segregation can exert sex ratio skews, indicating that these hormones may act directly on the germinal disc to influence which sex chromosome is retained in the oocyte and which is discarded into an unfertilizable polar body. We aimed to uncover the gene pathways involved in this process by testing whether treatments with testosterone or corticosterone that were previously shown to influence sex ratios elicit changes in the expression of genes and/or gene pathways involved in the process of meiotic segregation. We injected laying hens with testosterone, corticosterone, or control oil 5h prior to ovulation and collected germinal discs from the F1 preovulatory follicle in each hen 1.5h after injection. We used RNA-sequencing (RNA-seq) followed by DESeq2 and gene set enrichment analyses to identify genes and gene pathways that were differentially expressed between germinal discs of control and hormone-treated hens. Corticosterone treatment triggered downregulation of 13 individual genes, as well as enrichment of gene sets related to meiotic spindle organization and chromosome segregation, and additional gene sets that function in ion transport. Testosterone treatment triggered upregulation of one gene, and enrichment of one gene set that functions in nuclear chromosome segregation. This work indicates that corticosterone can be a potent regulator of meiotic processes and provides potential gene targets on which corticosterone and/or testosterone may act to influence offspring sex ratios in birds.

been uploaded to Gene Expression Omnibus
(GSE131363).

**Funding:** MTM and KJN Award #1456442
(collaborative grant to KJN and MTM) National
Science Foundation https://www.nsf.gov/ The
funders had no role in study design, data collection
and analysis, decision to publish, or preparation of
the manuscript.

**Competing interests:** The authors have declared
that no competing interests exist.

## Introduction

Hundreds of scientific observations since the early 1900's suggest that female birds are able to bias the sexes of the offspring they produce in response to external cues [1]. Birds bias primary offspring sex ratios in response to a variety of environmental conditions, such as territory quality [2,3], food availability [4,5], and laying date [6,7]. Both mate attractiveness [8,9] and maternal condition [10,11] also exert a significant influence on offspring sex ratios. In most cases, sex ratio manipulation happens without the loss of eggs or embryos in the laying sequence, which suggests that females can alter sex ratios prior to ovulation [12]. In birds, females are the heterogametic sex, contributing either a male-producing Z chromosome or a female-producing W chromosome. Segregation of these sex chromosomes within the germinal disc (GD), in which one is pulled into the polar body to be discarded while the other is kept in the oocyte, occurs just prior to ovulation [1,13,14]. A body of research suggests that hormones are likely mediators of sex ratio adjustment during this time because they can transduce environmental and social experiences into physiological responses, and treatment with multiple hormones has biased offspring sex ratios in birds [15].

Most mechanistic studies of sex ratio manipulation have focused on two hormones, testosterone and corticosterone, because they are necessary for normal reproductive functioning and elevations of both occur right before ovulation [16]. Females with elevated testosterone levels over a long period of time bias their offspring sex ratios towards males [17–20]. Rutkowska and Cichon [19] showed that female zebra finches (*Taeniopygia guttata*) injected with testosterone when clutches were initiated produced significantly more sons later in the clutch. Testosterone administered just prior to sex chromosome segregation has also resulted in male-biased sex ratios [21]. Corticosterone also triggers sex ratio biases, but the direction of the skew depends on the timing of exposure. Chronic elevations of corticosterone skew sex ratios towards a female bias [18,22,23] and laying hens that receive corticosterone treatment 4h pre-ovulation produce significantly more daughters [24]. However, a dose of corticosterone administered 5h prior to meiotic segregation results in male-biased sex ratios in laying hens and zebra finches [25,26]. In addition, Aslam et al. [27] showed that corticosterone treatment influenced the relationship between body mass and sex ratios of unincubated eggs. These studies illustrate that treatment with testosterone and corticosterone can influence sex ratios of avian offspring. Further, treating birds with these hormones just hours prior to ovulation of a follicle can alter the sex chromosome that the ovulated follicle receives.

There are several proposed mechanisms to explain how hormones may act to influence the sex of the ovulated oocyte [1,13]. There is evidence that centromere centrality on the chromosome (i.e., where spindle fibers attach to pull chromosomes apart) can influence which chromosome is retained in the egg and which is shuttled to the polar body [28,29]. The size and position of centromeres in birds is particularly plastic and differs between the W and Z chromosome, because the W chromosome contains large telomeres [30], which could potentially influence the centrality of its centromere [1]. Sex steroids and glucocorticoids can influence telomere lengths [31,32,33], making the alteration of telomere lengths a potential mechanism underlying sex ratio adjustment in birds, though in these cases telomere lengths changed over a period of days or weeks. Thus it is unclear whether telomere lengths can be altered within hours, as would be necessary for peri-ovulatory sex ratio adjustment.

The W chromosome also contains far less negatively charged DNA compared to the Z chromosome [34]. Spindle fiber movement (via microtubules) is influenced by charged ion gradients and hormone-induced charge gradients within the GD could result in differential movement of the sex chromosomes and/or the entire spindle apparatus [13]. Several other factors, like centromere and kinetochore size, and epigenetic modifications, may also impact the

number of spindle fiber attachments on the chromosome, either directly [35,36] or by altering chromosome compaction [37]. The "centromere drive" hypothesis suggests that the number of microtubule attachments may influence which chromosomes are retained in the egg ([38], reviewed in [13]). Finally, spindle function could underlie differential retention of the sex chromosomes; work in horses, marsupials, and frogs showed that the spindle apparatus rotates just prior to anaphase I to determine the ultimate fate of the chromosomes [39,40,41].

Our aim was to examine how steroids (testosterone and corticosterone) affect the genomic processes that underlie sex ratio adjustments in domestic chickens *(Gallus gallus)*. We injected laying hens subcutaneously with either a control, testosterone, or corticosterone solution just before meiotic segregation and collected GDs from the F1 follicles during sex chromosome segregation for RNA-seq analyses. We predicted that treatment with these hormones would alter the expression of genes involved in the process of sex chromosome segregation, specifically those involved in chromosome alignment, spindle-fiber formation, chromosome attachment, and chromosome movement towards the poles of the oocyte. Based on the hypothesized mechanisms presented above, we expect to see differences in the expression of genes controlling size/location of the centromere (e.g., via telomere lengths), ionic gradients, epigenetic modifications, and spindle attachments or function. The overall idea is that changes in the expression of key genes in combination with key differences between the W and Z chromosomes may influence which sex chromosome ends up in the egg, and which is sent to an unfertilizable polar body. To the best of our knowledge, this study is the first to test for an effect of steroid hormones on the differential segregation of sex chromosomes at the molecular level during meiosis.

## Methods

### Animal husbandry and experimental timing

Single-comb Hy-Line W36 White Leghorn hens (n = 400) were reared on the floor according to Hy-Line guidelines until they reached reproductive maturity. We then transferred them to individual layer cages in a single room, where they had *ad libitum* access to food and water throughout this study. They were kept on a standard breeding light schedule of 16h light: 8h dark. In laying hens, ovulation occurs 15–30 minutes after oviposition of the previous egg [14]. We could predict the timing of ovulation of each individual bird used in the study by recording their egg-laying patterns. When hens reached 30 weeks of age (after egg production has maximized), we monitored hens between 0700 AM– 1000 AM daily for 5 weeks, and determined the precise time that the egg was laid based on the rate of egg cooling using thermal imaging using a (Flir One Pro, Flir® Systems, Wilsonville, OR) as described in [42].

### Hormone pilot study

Prior to the start of the experiment, we did a pilot study to validate that our hormone injections were effective in raising hormone plasma levels for several hours after being administered. We subcutaneously injected 21 consistent layers between 0400AM-0500AM, which was the same timing planned for the full experiment. These birds were from the same flock used in our subsequent experiment, which is important given that the rates of hormone metabolism have been shown to vary among flocks [24]. The 21 birds were randomly split into 3 different treatments: (1) 7 birds received a control injection (0.5ml of peanut oil) (2) 7 birds received a testosterone injection (1.5mg testosterone dissolved in 0.5ml of peanut oil), and (3) 7 birds received a corticosterone injection (1.5mg of corticosterone dissolved in 0.5ml of peanut oil). Baseline blood (1.0ml) was collected just prior to giving the injection via venipuncture of the brachial vein. Blood was again collected at both 1h and 2h post-injection. All samples were

kept on ice until transported back to the lab. Plasma was collected from the samples and stored at -80˚C.

For the hormone extraction procedure, 20ul of each sample was pipetted into separate glass vials, and 3ml of diethyl ether was added to each vial. The vials were vortexed for 30 seconds, then placed into a centrifuge and spun at 4˚C at 1800 rpm for 9 minutes. The samples were then kept at -80˚C for 7 minutes. Following this step, the samples were thawed and the supernatant was immediately decanted into clean glass culture tubes and dried overnight. Hormone plasma levels were determined using a corticosterone ELISA kit (item no. 501320, Cayman Chemical) and a testosterone ELISA kit (ADI-901-065, Enzo Life Sciences). The extracted samples were suspended in assay buffer and 50ul of each sample was added to the wells for both kits used. The assays were conducted following the protocol given with each kit. Results were analyzed using a repeated measures ANOVA for each hormone using Statview© software (SAS institute, Cary, NC, USA).

## Hormone administration and tissue collection

Approximately 3–5 h before ovulation, the attachment of spindle fibers and segregation of sex chromosomes in the GD occurs [14], thus injections were timed for 5h prior to ovulation. The 165 most consistent layers were randomly assigned to receive one of the three treatments described above (testosterone, corticosterone, or control). At approximately 1.5h post-injection (which is 3.5h before ovulation), when the sex chromosomes should be segregating, all hens were killed via injection of 1ml Euthasol and the F1 follicles were collected. The GD region was excised from the F1 follicle, briefly washed in Krebs buffer to remove any yolk material and placed in 0.5mL reaction tube filled with 100μl of Buffer RL (Norgen Biotek Corp.). Then, 100μl of 70% ethanol was added to the sample and it was vortexed thoroughly until the GD material was completely broken down. The samples were placed in -80˚C for future use. Eight samples were lost during sample collection due to the difficulty of excising the germinal disc, leaving a total of 157 valid GD samples collected across eight days between 18 April 2018 and 8 May 2018. Of these, we selected the 45 samples for each treatment with the best RNA yields after extraction, leaving 45 samples collected from testosterone-treated birds, 45 samples from corticosterone-treated birds, and 45 samples from the control birds.

## RNA isolation, pooling, and quality assessment

The RNA from the samples was extracted using the Norgen's Single Cell RNA Purification Kit (catalog no. 51800, Norgen Biotek Corp.). A Nanodrop© spectrophotometer (ND-1000 Nanodrop Technologies, Wilmington, DE, USA) was used to assess A260/A280 ratios for determination of initial RNA purity and concentration of each sample. To ensure adequate RNA quantity for analysis, we pooled nine GD region samples (8μl per replicate), ultimately creating 5 pooled replicates per treatment group. Final RNA integrity of each replicate was assessed on an Agilent 2100 Bioanalyzer instrument (Model 62939B, Agilent Technologies, Santa Clara, CA, USA) at the Georgia Genomics and Bioinformatics Core (GGBC) at the University of Georgia (Athens, GA).

## RNA-seq library preparation and sequencing

The GGBC at the University of Georgia conducted all library preparation and sequencing. The KAPA Stranded mRNA-Seq kit was used for the construction of NGS stranded RNA library for each replicate (KK8421, KAPA Biosystems, Wilmington, MA, USA). All libraries were pooled together by qPCR using the Roche LightCycler 480 II (product no. 05015278001, Roche Molecular Systems, Inc., Pleasanton, CA, USA). For this step, the KAPA Library

Quantification kit (Illumina) with qPCR Master Mix optimized for LightCycler 480 was used (KK4854, KAPA Biosystems, Wilmington, MA). Next, the pooled library underwent pre-sequencing quality control. The DNA concentration of the pooled library was quantified using the Qubit HS dsDNA assay (catalog no. Q32854, ThermoFisher Scientific, Waltham, MA, USA). The Fragment Analyzer Automated CE System (Advanced Analytical Technologies, Ankeny, IA, USA) was used to visual the size distribution of the library. Then, qPCR was performed using the same kit and PCR products were quantified as described above. The pooled libraries were sequenced on an Illumina NextSeq 500 for 150 cycles with a PE75 high output flow cell.

### RNA-sequencing data processing and analysis

Trimmomatic software, version 0.36, [43] was employed to remove adapters and quality trim raw reads; reads with a trimmed length below a 35-base threshold were discarded. Reads were assessed for quality using FastQC both before and after trimming [44]. Trimmed reads were aligned to the *Gallus gallus* genome (Ensemble Gallus_gallus-5.0, GCA_000002315.3) using Tophat2 [45] run at default settings and BAM files were name sorted prior to extraction of raw gene counts with HTSeq [46] for generation of the count matrix. The matrix contained 24,881 rows (genes) and after removal of rows containing low counts (<10 reads) this was reduced to 17,319 rows. All analyses were carried out in R (version 3.4.3; R Development Core Team). The Bioconductor package *DESeq2* [47] was used to identify differentially expressed genes (DEGs) and calculate normalized gene counts. Pairwise comparisons between the corticosterone-treated GDs or testosterone-treated GDs vs control GDs were made and DEGs with a false discovery rate (FDR) $\leq$ 0.05 were considered statistically differentially expressed. The web-based Biological Database Network (bioDBnet) conversion utility [48] was used to extract gene symbols for the Ensembl gene IDs.

### Gene set enrichment analyses

In order to better understand the potential roles played by up- and down-regulated genes from each hormone treatment, we performed a Gene Set Enrichment Analysis (GSEA) for each pairwise comparison [49] using the *fgsea* package in R [50]. The gene lists used were comprised of all identified genes ranked by $-\log_{10}P^*$sign (logFC) for each comparison. The gene sets used included the curated Kyoto Encyclopedia of Genes and Genomes (KEGG) (c2) and gene ontology (GO) biological process (c5) from the Molecular Signature Database (MSigDBv2.5, [51]). The significance was assessed with 10,000 permutations, with a significance cut-off set as a Benjamini-Hochberg adjusted p $\leq$ 0.05 [52]. Significant gene sets were visualized using Enrichment Map (nodes represent gene sets and edges represent mutual overlap) to cluster highly redundant gene sets into functional groups, which we defined using AutoAnnotate in Cytoscape (v3.6.1; [53]). We only show functional groups containing 5 or more gene sets.

## Results

### Hormone pilot study

Results from the hormone pilot study indicate that the injections of testosterone and corticosterone were effective in raising the respective hormone concentrations in the hens (Fig 1). Testosterone levels were significantly elevated above controls when compared to baseline concentrations at 1h and 2h post-injection (Treatment: $F_{1,18} = 89.33$, p < 0.001, Time: $F_{2,18} = 72.48$, p < 0.001) as was corticosterone (Treatment: $F_{1,18} = 6.061$, p < 0.04, Time: $F_{2,16} = 4.68$, p = 0.02).

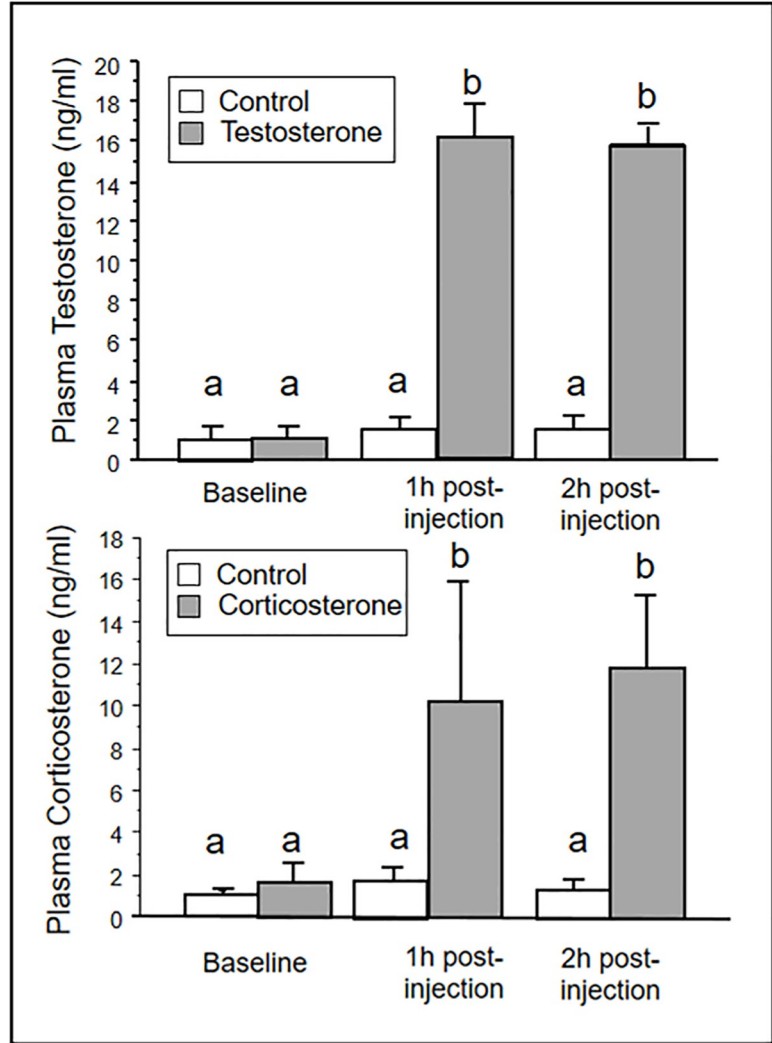

**Fig 1. Testosterone and corticosterone concentrations in plasma before and after an injection with testosterone (n = 7, 1.5mg in 0.5ml peanut oil), corticosterone (n = 7, 1.5mg in 0.5ml peanut oil), or control oil (n = 7, 0.5ml peanut oil).** Letters above bars indicate significant differences. Bars indicate means ± SE.

### Differentially Expressed Genes (DEGs) in response to hormone treatment

We tested comparisons between corticosterone-treated versus control birds, and testosterone-treated versus control birds. An average of ca. 42 million trimmed, PE reads/sample were mapped to the *G. gallus* genome using Tophat2 with an average overall alignment rate of 93% and an average concordant alignment rate > 88%. Only one gene, iodothyronine deiodinase 2 (DIO2), was differentially expressed between testosterone and control samples; this gene was upregulated in the samples from testosterone-treated hens (S1 Table). Thirteen DEGs were identified in the corticosterone-treated versus control group (Fig 2). Five of these genes, CTD small phosphatase like (CTDSPL) (also called SCP3), exostosin glycosyltransferase 2 (EXT2), leukemia inhibitory factor receptor alpha (LIFR), transcription factor 12 (TCF12), and SET domain containing lysine methyltransferase 7 (SETD7), are either directly involved in the process of oocyte maturation, have the potential to influence the ionic gradient within the GD, or can exert epigenetic modifications on the chromosomes. For the full list of DEGs see S1 Table.

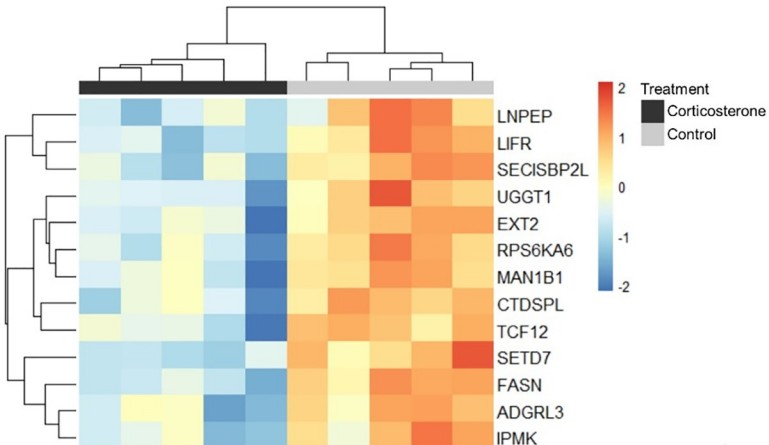

**Fig 2. Heatmap depicting differentially expressed genes between control (light gray) and corticosterone (dark gray) treatments.** Each column is a set of 9 pooled samples, and each row is a gene. Heatmaps are scaled across rows to allow for comparisons of gene expression across individuals. Color indicates log-transformed normalized counts; blue indicates relatively lower expression and red indicates relatively higher expression. Rows and columns are clustered using Euclidean distance.

## Gene set enrichment analyses

Next, we used GSEA to identify sets of genes that are regulated in GDs from testosterone- and corticosterone-treated hens. For testosterone-treated hens, 59 gene sets (56 GO and 3 KEGG) were significantly enriched with genes in a positive direction (upregulated) in testosterone-treated hens relative to controls and 17 gene sets (16 GO and 1 KEGG) were enriched with genes in a negative direction (downregulated; full GSEA results can be found in S2 Table). We visualized gene sets relevant for sex determination using the Enrichment Map plug-in for Cytoscape. Within the upregulated gene sets, five overlapped to form a functional group related to cell cycle regulation (Fig 3A), one of which included genes directly related to nuclear chromosome segregation (Fig 3B). No relevant functional groups were found in downregulated gene sets.

For corticosterone-treated hens, 86 gene sets (80 GO and 6 KEGG) were significantly upregulated in corticosterone-treated hens relative to controls and 87 gene sets (81 GO and 6 KEGG) were downregulated (full GSEA results can be found in S3 Table). Within upregulated gene sets, several gene sets overlapped to form function groups related to chromosome segregation, spindle organization, cell cycle, chromatin modification, and DNA repair (Fig 4A). Specifically, genes involved in chromosome segregation, regulation of microtubule-based processes, and telomere maintenance via recombination were positively related to corticosterone treatment (Fig 4A). Within downregulated genes sets, 17 gene sets overlapped to form a functional group related to ion transport (Fig 3A), one of which included genes directly involved in anion transport (Fig 4B).

## Discussion

Both corticosterone and testosterone treatment, given at the time sex chromosomes segregate in hens, triggered significant effects on both expression of individual genes and of gene networks in the preovulatory GD. Testosterone's influences appeared to be less dramatic than those triggered by corticosterone treatment. Our results showed only one DEG between hens treated with testosterone 5h prior to ovulation and hens that received a control injection; This

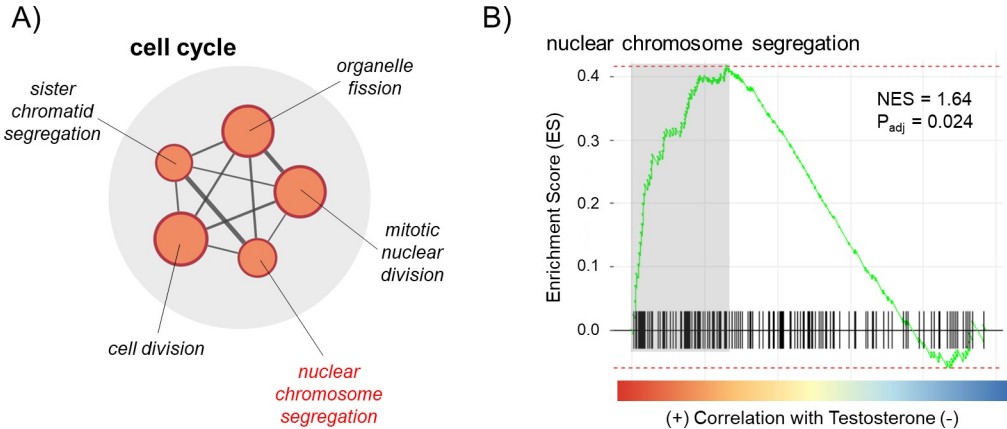

**Fig 3.** (A) Enrichment Map showing a functional group of gene sets related to cell cycle that were detected in a Gene Set Enrichment Analysis (GSEA) comparing testosterone- versus control-treated hens, one of which was nuclear chromosome segregation (highlighted in red). Only gene sets with $p_{adj} < 0.05$ are included. Node size corresponds to gene set size. Normally, color would designate the direction the gene sets were altered (red = enriched in testosterone-treatment, blue = depleted in testosterone-treatment), but in this case, all were enriched in samples from testosterone-treated hens, so all are red. Line thickness connecting the gene set nodes represents the degree of gene overlap between the two sets. (B) GSEA enrichment plot for a select gene set. Black bars represent the position of members of the gene set in the ranked list together with the running enrichment score (ES; plotted in green). The adjusted p-value and normalized enrichment score (NES) are presented. Leading edge genes (i.e., those that contribute most to the ES) are highlighted in gray and can be found in S2 Table.

gene (DIO2) was upregulated in hens treated with testosterone, but it is not known to exhibit functions that would likely influence sex chromosome movement. Corticosterone treatment, on the other hand, triggered downregulation of 13 DEGs, 3 of which are particularly interesting in terms of how they may play into an existing hypothesized mechanism for how sex ratio adjustment occurs. Furthermore, both testosterone and corticosterone treatment resulted in enriched gene sets related to sex ratio adjustment. Altogether, these data suggest exposure to steroids, particularly corticosterone, has the potential to influence many of the mechanisms underlying sex determination, which we discuss in detail below.

Corticosterone treatment had the strongest effect, leading to the downregulation of several DEGs with the potential to adjust sex ratios based on 4 hypothesized mechanisms (Fig 5). EXT2 encodes a protein involved in the biosynthesis of heparin sulfate, a molecule located on the cell surface and known for its extreme negative charge due to sulfate and carboxyl groups [54,55]. Spindle function and movement of chromosomes within the cell is controlled by nanoscale electrostatic forces [56]. Perhaps if the biosynthesis of heparin sulfate is reduced, the cell surface would be less negatively charged. Since the Z chromosome contains more negatively charged DNA compared to the W [34], this could potentially make the Z chromosome more likely to migrate towards the surface of the oocyte (and thus the polar body) (hypothesis 2, Fig 5). This would result in more female offspring, as has been shown to occur as a result of both chronic exposure to and low-dose acute treatment with corticosterone [23,24].

CTDSPL (also called SCP3) encodes for a protein that is part of the synaptonemal complex, which is a meiosis-specific structure that is responsible for the pairing of the homologous chromosomes [57]. Prior work suggests this gene is sensitive to hormone levels and lower expression is associated with the disruption of meiotic progression [58]. It is possible that downregulating this gene could lead to fewer links between homologous chromosomes, potentially altering the ability of spindle fibers to attach (hypothesis 3, Fig 5).

Finally, SETD7 encodes for a histone-lysine methyltransferase that induces methylaton of histone 3 at lysine 4 (H3K4) [59]. In humans and mice, demethylation at this site is thought to

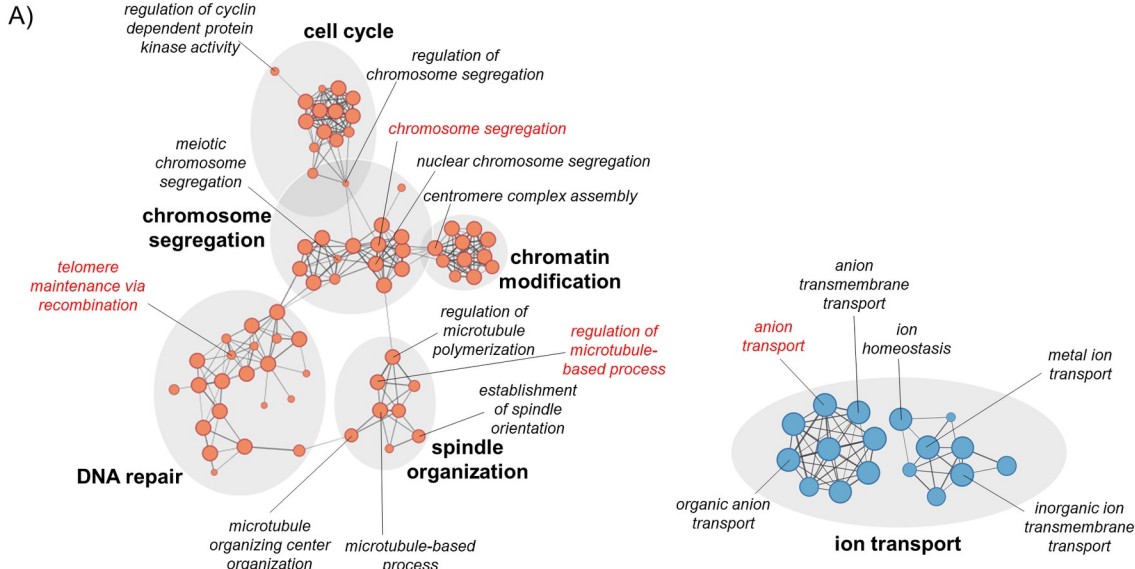

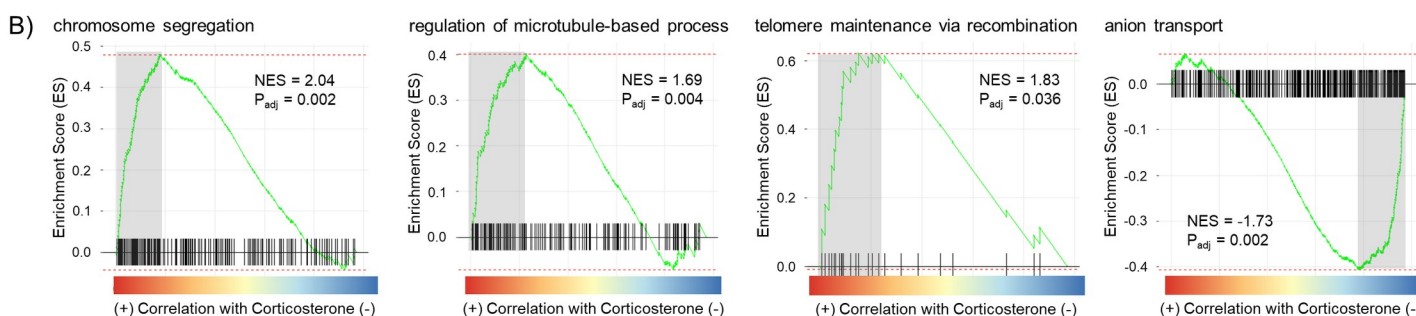

**Fig 4.** (A) Enrichment Map showing functional groups (shaded in gray) of gene sets detected in a Gene Set Enrichment Analysis (GSEA) comparing corticosterone-versus control-treated hens; select gene sets are highlighted in red. Only functional groups containing $\geq 5$ gene sets with $p_{adj} < 0.05$ are included. Node size corresponds to gene set size and color designates the direction the gene sets were altered (red = enriched in corticosterone-treatment, blue = depleted in corticosterone-treatment). Line thickness connecting the gene set nodes represents the degree of gene overlap between the two sets. (B) GSEA enrichment plot for select gene sets. Black bars represent the position of members of the gene set in the ranked list together with the running enrichment score (ES; plotted in green). The adjusted p-value and normalized enrichment score (NES) are presented. Leading edge genes (i.e., those that contribute most to the ES) are highlighted in gray can be found in S2 Table.

contribute to physical organization of the centromere [60], a factor that is critically important in the connection of spindle fibers. Whether the activity of this methyltransferase can differentially influence the histones in the W versus Z chromosome remains unclear. However methylation at this site also blocks nucleotide-based methylation at CpG sites [61]. The W and Z chromosomes are differentially sensitive to nucleotide-based methylation, because the Z chromosome contains a 460-kb long region on its short arm where CpG sites are highly methylated during development [62]. Work in plants shows that the level of DNA methylation can influence segregation distortion [63], though we could not find published tests of this in animal systems. If methylation also controls chromosome movement in animals, this could provide a path by which SETD7 may be involved in directing how the two chromosomes migrate during meiotic segregation thereby skewing sex ratios of birds (hypothesis 3, Fig 5). Of the remaining DEGs, two are known to function in the process of oocyte maturation. LIFR participates in the LIF/STAT3 pathway, which is important in oocyte maturation [64]. FASN is involved in lipogenic activity of oocytes [65] and was also previously shown to be sensitive to glucocorticoids in mammals [66]. While more work needs to be done to determine whether the activity of

**Fig 5. Differentially Expressed Genes (DEGs) (all were downregulated) and significantly enriched GO terms in corticosterone compared to control-treated birds that may function in each of the proposed hypotheses for sex ratio adjustment in birds.** Plus and minus signs indicate the direction of the correlation with corticosterone treatment.

these genes influences W and Z chromosomal segregation, and whether they directly influence sex determination, the current study suggests they may be involved in mechanisms by which steroids influence sex ratios.

While examining expression patterns of individual genes can yield important results, genes do not act on their own, but instead act in concert within a network of genes. GSEA results indicated that both testosterone and corticosterone treatments triggered enrichment of gene sets that are likely important in the process of sex determination. In corticosterone-treated hens, the gene sets that function in telomere maintenance via recombination and centromere complex assembly were both upregulated and could contribute to the first hypothesis concerning centromere centrality (Fig 5) [1]. Downregulated gene sets in corticosterone-treated hens overlapped to form a functional group related to ion transport, the function of which could potentially alter the electrochemical gradient in the GD (hypothesis 2, Fig 5). We also found functional groups of upregulated gene sets related to spindle organization, which could influence the number of spindle attachments and determine which sex chromosome is retained (hypothesis 3, Fig 5). Finally, upregulated gene sets, including chromosome segregation and establishment of spindle orientation, support the spindle flip hypothesis (hypothesis 4, Fig 5), where the spindle flips just before ovulation to control which set of chromosomes are retained

in the egg [39–41]. Interestingly, chromosome segregation and spindle are two gene sets that were also enriched in hens that produce female-biased eggs [67]. Hypothesis 2 and 3 received the most support suggesting corticosterone may primarily skew sex ratios through manipulation of the ion gradient and/or spindle attachments.

Testosterone-treated GDs had fewer enriched gene sets that were relevant to the process of meiosis, though there was a functional group of upregulated gene sets controlling cell cycle processes. Perhaps the most interesting enriched gene set in testosterone-treated hens is nuclear chromosome segregation, which was also upregulated in corticosterone-treated hens. Given that both testosterone and corticosterone trigger skews in offspring sex ratios, perhaps they both do so through modifications of genes found in this particular gene set. It is difficult to explain why the two hormones were correlated with the gene sets in the same direction, given that they generally exert opposite effects on sex ratios. However, like testosterone treatment, acute treatment with corticosterone has elicited male-biased sex ratios in previous work [25,26], suggesting genes involved in nuclear chromosome segregation are worth exploring further. We thought that an elevation of corticosterone in response to testosterone treatment may explain this, but after quantifying corticosterone in plasma one hour after testosterone injection, we saw no such elevation (data not shown).

The data presented herein provide mounting evidence that environmental variables influence sex ratios through the actions of hormones, particularly corticosterone. We additionally present some functionally relevant candidate genes and gene sets that can be further tested for influences on the process of sex ratio adjustment. It is particularly important for future work to consider how the W and Z chromosomes may be differentially sensitive to the functions of these gene sets to better elucidate how their final orientation and/or movement may be influenced. For example, given potential charge differences between the two chromosomes (Z contains more negatively charged DNA compared to the W [34], it is relatively easy to see how processes that could control ion gradients within the GD could ultimately determine position of the sex chromosomes. If there are differences in centromere size between the two chromosomes, which has not yet been tested, altering processes that influence microtubule and/or protein attachments to the centromere could disproportionately influence one sex chromosome over another. The W chromosome does contain a mega-telomere not found in the Z chromosome [30], suggesting the gene set related to telomere maintenance could exert a larger effect on centromere centrality on the W chromosome, ultimately influencing the direction in which it is pulled. Future studies should also test whether the electrochemical gradient within the GD corresponds to retention of a particular sex chromosome, and whether silencing genes in the identified pathways influences the direction of sex chromosome movement.

To date, knowledge about the process of meiotic segregation in birds is extremely limited. Our data provide insight not only into potential mechanisms of sex ratio adjustment, but also into gene pathways that may be involved in meiosis in birds. It is now possible to compare these findings with what is already known about this process in other systems to determine whether we can use mechanisms from other systems to further our understanding of meiotic processes in birds.

## Supporting information

**S1 Checklist.**
(PDF)

**S1 Table. Hormone treatment validation data.**
(XLSX)

**S2 Table. Differentially expressed genes between testosterone- and control-treated hens and between corticosterone- and control-treated hens.** Values are relative to birds from hormone treatments.
(XLSX)

**S3 Table. Results from a Gene Set Enrichment Analysis (GSEA) for testosterone vs control samples.** Size indicates the number of genes in both the gene set and the expression dataset. The enrichment score (ES) reflects the degree to which a gene set is overrepresented in a list of genes. The normalized enrichment score (NES) accounts for differences in gene set size; this is the primary result of the GSEA. Leading edge genes drive the enrichment score. P values are adjusted using a Benjamini-Hochberg correction.
(XLSX)

**S4 Table. Results from a Gene Set Enrichment Analysis (GSEA) for corticosterone vs control samples.** Size indicates the number of genes in both the gene set and the expression dataset. The enrichment score (ES) reflects the degree to which a gene set is overrepresented in a list of genes. The normalized enrichment score (NES) accounts for differences in gene set size; this is the primary result of the GSEA. Leading edge genes drive the enrichment score. P values are adjusted using a Benjamini-Hochberg correction.
(XLSX)

## Acknowledgments

We appreciate Sergio Alcantar and Alyson Ming Wright for helping with oviposition monitoring. We thank Caroline R. Cummings and James E. Curry with tissue collections. Bioinformatics analyses of RNA-seq data and gene set analyses generated in this project was performed with the help of consultants in the University of Georgia's Georgia Genomics & Bioinformatics Core (GGBC). Computational work was done using the high-performance computing resources at the Georgia Advanced Computing Resource Center (GACRC).

## Author Contributions

**Conceptualization:** Elizabeth R. Wrobel, W. Walter Lorenz, Mary T. Mendonça, Kristen J. Navara.

**Data curation:** Elizabeth R. Wrobel, Alexandra B. Bentz, W. Walter Lorenz, Mary T. Mendonça, Kristen J. Navara.

**Formal analysis:** Elizabeth R. Wrobel, Alexandra B. Bentz, W. Walter Lorenz, Stephen T. Gardner, Mary T. Mendonça, Kristen J. Navara.

**Funding acquisition:** Mary T. Mendonça, Kristen J. Navara.

**Investigation:** Elizabeth R. Wrobel, Mary T. Mendonça, Kristen J. Navara.

**Methodology:** Elizabeth R. Wrobel, Stephen T. Gardner, Mary T. Mendonça, Kristen J. Navara.

**Project administration:** Mary T. Mendonça, Kristen J. Navara.

**Resources:** Mary T. Mendonça, Kristen J. Navara.

**Supervision:** Mary T. Mendonça, Kristen J. Navara.

**Validation:** Elizabeth R. Wrobel, Kristen J. Navara.

**Writing – original draft:** Elizabeth R. Wrobel, Alexandra B. Bentz, Mary T. Mendonça, Kristen J. Navara.

**Writing – review & editing:** Elizabeth R. Wrobel, Alexandra B. Bentz, W. Walter Lorenz, Stephen T. Gardner, Mary T. Mendonça, Kristen J. Navara.

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
