## [Decision Letter · Decision Letter 0]

14 Jan 2020

PONE-D-19-33739

Corticosterone and testosterone treatment influence expression of gene pathways linked to meiotic segregation in preovulatory follicles of the domestic hen

PLOS ONE

Dear Dr. Navara,

Thank you for submitting your manuscript to PLOS ONE. After careful consideration, we feel that it has merit but does not fully meet PLOS ONE’s publication criteria as it currently stands. Therefore, we invite you to submit a revised version of the manuscript that addresses the points raised during the review process.

*Minor revisions are necessary to improve the paper.  If you like to revise the paper in accordance with each of these comments, please create an "author response" file with a point-by-point response to each comment, clearly describing how they have been addressed in the revision*.

We would appreciate receiving your revised manuscript by Feb 28 2020 11:59PM. To enhance the reproducibility of your results, we recommend that if applicable you deposit your laboratory protocols in protocols.io, where a protocol can be assigned its own identifier (DOI) such that it can be cited independently in the future. For instructions see: http://journals.plos.org/plosone/s/submission-guidelines#loc-laboratory-protocols

We look forward to receiving your revised manuscript.

Kind regards,

Aneta Agnieszka Koronowicz, PhD

Academic Editor

PLOS ONE

Journal Requirements:

2.  In your Methods section, please provide additional details regarding the animals used in your study and ensure you have described the source. For more information regarding PLOS' policy on materials sharing and reporting, see " ext-link-type="uri" xlink:type="simple">https://journals.plos.org/plosone/s/materials-and-software-sharing#loc-sharing-materials."

3. As part of your revision, please complete and submit a copy of the ARRIVE Guidelines checklist, a document that aims to improve experimental reporting and reproducibility of animal studies for purposes of post-publication data analysis and reproducibility: https://www.nc3rs.org.uk/arrive-guidelines. Please include your completed checklist as a Supporting Information file. Note that if your paper is accepted for publication, this checklist will be published as part of your article.

4. In your Methods section, please include a comment about the state of the animals following this research. Were they euthanized or housed for use in further research? If any animals were sacrificed by the authors, please include the method of euthanasia and describe any efforts that were undertaken to reduce animal suffering.

5. We note that you are reporting an analysis of a microarray, next-generation sequencing, or deep sequencing data set. PLOS requires that authors comply with field-specific standards for preparation, recording, and deposition of data in repositories appropriate to their field. Please upload these data to a stable, public repository (such as ArrayExpress, Gene Expression Omnibus (GEO), DNA Data Bank of Japan (DDBJ), NCBI GenBank, NCBI Sequence Read Archive, or EMBL Nucleotide Sequence Database (ENA)). In your revised cover letter, please provide the relevant accession numbers that may be used to access these data. For a full list of recommended repositories, see http://journals.plos.org/plosone/s/data-availability#loc-omics or http://journals.plos.org/plosone/s/data-availability#loc-sequencing.

Additional Editor Comments (if provided):

Reviewers' comments:

Reviewer's Responses to Questions

**Comments to the Author**

1. Is the manuscript technically sound, and do the data support the conclusions?

Reviewer #1: Yes

Reviewer #2: Yes

2. Has the statistical analysis been performed appropriately and rigorously? 

Reviewer #1: Yes

Reviewer #2: Yes

3. Have the authors made all data underlying the findings in their manuscript fully available?

Reviewer #1: Yes

Reviewer #2: No

4. Is the manuscript presented in an intelligible fashion and written in standard English?

Reviewer #1: Yes

Reviewer #2: Yes

5. Review Comments to the Author

Reviewer #1: This genomic study has been properly designed and performed. I do not have any remarks concerning methodology of the study including genomic tools and statistics used. However, since the small number of the studies in this area I have few comments to the general idea of the study.

1. Authors investigate the influence of two hormones on sex selection in hen. One of these hormones, namely corticosterone, plays a role of stress hormone in birds (as cortisol does in mammals). So maybe the hypotheses of sex selection given in the paper should also include the idea that stress factors (resulting in corticosterone release) bias offspring sex ration in female birds. If there is no stress (no corticosterone release) the standard “physiological setting” of sex selection is male-directed. This could explain obtained data that testosterone treatment resulted in significant regulation of only one gene. This is surprising since the testosterone administration resulted in significant increase of plasma testosterone level but the gene expression in GD seem to be unaffected. Bearing in mind the number of genes investigated in the study the number of differentially regulated genes by testosterone is quite small.

2. One of the drawbacks of this paper is lack of clinical evaluation of treatments applied. In my opinion part of the birds from each experimental group should be kept further for the evaluation of male and female progeny ratio produced after treatment. Therefore few questions arises:

• We have here the data shoving effectiveness of hormones administration (increased plasma level of both hormones after administration – Fig 1) but we do not know whether i.e. corticosterone administration influenced also endogenous testosterone and vice versa.

• Moreover, we do not know whether these treatment in fact changed male or female embryos ratio, so the study evaluates only the influence of the administration of these two hormones on gene expression in GD of F1 follicles but not on offspring sex selection.

• Data cited by the Authors suggest the influence of time of hormones administration before ovulation on the offspring sex selection. As I assume by administering the hormone roughly 5 hours before ovulation the Authors expected the skew toward male offspring. But why they decide to select the timepoint 5 hours before ovulation and not i.e. 3 hours before ovulation.

3. Why Authors did not validate obtained results by Real-time PCR and were not trying to measure the level of proteins encoded by regulated genes. There are some suggestions in the paper that post-translational modifications could play here the important role, so the question arises whether the pure genomic approach is sufficient to give any hypotheses concerning sex selection.

4. Did Authors received in genomic analyses also the significantly regulated unknown genes? Since the process of sex determination in bird offspring is in fact not well understood maybe the key role in this process play genes (and proteins) not previously described (and not taken into consideration in ontology analyses by different software).

Reviewer #2: Offspring primary sex ratio bias in birds has been a very popular topic and the idea that outcome of meiotic division could be non-random inspires both basic and applied research. The study by Wrobel at al. is a step forward into investigations on how hormonal levels in the female just prior to ovulation could affect outcome of sex-determine meiotic division in birds. This is probably the second study that looks at gene expression in the germinal disk in avian pre-ovulatory oocytes (after Aslam et al 2015), and the first one that does it following manipulation of hormones.

Authors demonstrate that injections of hormones affect gene expression patters in the germinal disk, but do not study the outcome of meiosis in those disks, as this would probably be not technically possible. However, they also do not provide information on how accurate are testosterone and corticosterone manipulations in altering offspring primary sex ratios. Recent meta-analyses (Merkling et al. 2018 and Podmokla et al. 2018) show that such effects are rather weak and more likely to occur in response to testosterone manipulation. Thus, the current study provides some valuable new data, but leaves several questions still open.

The experimental design is appropriate and sample sizes are good. The pilot study reassures that hormonal injections are effective. Molecular analyses are carried out up the standards.

I have reviewed this manuscript before, and currently, I was happy to find that the presentation of the study has greatly improved. Few remaining issues are listed below.

Line 70 – Sex steroids and glucocorticoids can influence telomere length. Please specify the tissue in which the cited references studied telomere length.

Line 101-2- The current study does not look at the differential segregation of the sex chromosomes.

Line 225 – Please check degrees of freedom for the denominators.

Line 646 – Should be “selected”, this also refers also lines 653, 657.

Supplementary materials are incomplete – original data for plasma hormone levels following injection of the hens is not provided.

6. PLOS authors have the option to publish the peer review history of their article (what does this mean?). If published, this will include your full peer review and any attached files.

Reviewer #1: No

Reviewer #2: No

---

## [Author Response · Author response to Decision Letter 0]

19 Mar 2020

1. Authors investigate the influence of two hormones on sex selection in hen. One of these hormones, namely corticosterone, plays a role of stress hormone in birds (as cortisol does in mammals). So maybe the hypotheses of sex selection given in the paper should also include the idea that stress factors (resulting in corticosterone release) bias offspring sex ration in female birds. If there is no stress (no corticosterone release) the standard “physiological setting” of sex selection is male-directed. This could explain obtained data that testosterone treatment resulted in significant regulation of only one gene. This is surprising since the testosterone administration resulted in significant increase of plasma testosterone level but the gene expression in GD seem to be unaffected. Bearing in mind the number of genes investigated in the study the number of differentially regulated genes by testosterone is quite small.

We thank the reviewer for this suggested hypothesis, and we now include it in the discussion (Lines 360-375). 

2. One of the drawbacks of this paper is lack of clinical evaluation of treatments applied. In my opinion part of the birds from each experimental group should be kept further for the evaluation of male and female progeny ratio produced after treatment. Therefore few questions arises:

• We have here the data shoving effectiveness of hormones administration (increased plasma level of both hormones after administration – Fig 1) but we do not know whether i.e. corticosterone administration influenced also endogenous testosterone and vice versa.

We agree that it would be nice to have these data. We no longer have the plasma from the pilot hormone validation portion of the study, however in future studies, it would be a good idea to see if the testosterone treatment influences hormone concentrations and vice versa. The fact that we saw quite different effects in response to corticosterone and testosterone treatments indicates, however, that the two hormones were not likely acting on the genes involved in sex ratio manipulation via effects on one another. We did add a discussion point on this (Lines 355-359).

• Moreover, we do not know whether these treatment in fact changed male or female embryos ratio, so the study evaluates only the influence of the administration of these two hormones on gene expression in GD of F1 follicles but not on offspring sex selection.

We agree that this is a limitation of the study, and when we were in the planning stages of the work, we brainstormed every possible way to get data on the sex ratio effects. However, we were limited by the number of birds that were laying at the exact intervals we needed, and we were not able to spare enough of them for a robust concurrent analyses of sex ratio effects. In the samples that we collected for gene expression analyses, it was impossible to determine sex because the chromosomes had not yet segregated. Despite this, we now know that these two hormones influence gene pathways related to meiotic segregation, and we can now take additional steps to manipulate these pathways and look at the outcomes on offspring sex. We added a discussion point on this (Lines 366-375).

• Data cited by the Authors suggest the influence of time of hormones administration before ovulation on the offspring sex selection. As I assume by administering the hormone roughly 5 hours before ovulation the Authors expected the skew toward male offspring. But why they decide to select the timepoint 5 hours before ovulation and not i.e. 3 hours before ovulation.

We chose 5h prior to ovulation because, according to previous research, meiotic segregation occurs 3-5h prior to ovulation. In addition, in our own previously published studies, we show that injection of these same amounts of these hormones at 5h prior to ovulation triggers sex ratio skews. So, we chose to use these exact same treatments to look at gene expression. We’ve attempted to clarify this in our methods section (Lines 149-151).

3. Why Authors did not validate obtained results by Real-time PCR and were not trying to measure the level of proteins encoded by regulated genes. There are some suggestions in the paper that post-translational modifications could play here the important role, so the question arises whether the pure genomic approach is sufficient to give any hypotheses concerning sex selection.

This is our first step. Due to the small amount of DNA that we get out of the germinal discs, we did not have enough to run both RNA seq and real-time quantitative PCR. In addition, because it was mostly gene networks that changed (rather than individual genes), this made it difficult to choose a subset to validate for this current paper. For future studies, we hope to perform several tests to validate the differences in the expression of the major gene networks presented in this paper. 

4. Did Authors received in genomic analyses also the significantly regulated unknown genes? Since the process of sex determination in bird offspring is in fact not well understood maybe the key role in this process play genes (and proteins) not previously described (and not taken into consideration in ontology analyses by different software).

Thanks for this suggestion. In the control-corticosterone comparison, there was only one differentially expressed gene that was unidentified, and in the control-testosterone comparison, there were none. We now indicate that there is an affected gene of unknown function in the presentation of results (Line 239), and also include a line in the discussion about the potential for that gene to have a function related to the process of sex determination (Lines 321-322). 

Reviewer #2: Offspring primary sex ratio bias in birds has been a very popular topic and the idea that outcome of meiotic division could be non-random inspires both basic and applied research. The study by Wrobel at al. is a step forward into investigations on how hormonal levels in the female just prior to ovulation could affect outcome of sex-determine meiotic division in birds. This is probably the second study that looks at gene expression in the germinal disk in avian pre-ovulatory oocytes (after Aslam et al 2015), and the first one that does it following manipulation of hormones.

Authors demonstrate that injections of hormones affect gene expression patters in the germinal disk, but do not study the outcome of meiosis in those disks, as this would probably be not technically possible. However, they also do not provide information on how accurate are testosterone and corticosterone manipulations in altering offspring primary sex ratios. Recent meta-analyses (Merkling et al. 2018 and Podmokla et al. 2018) show that such effects are rather weak and more likely to occur in response to testosterone manipulation. Thus, the current study provides some valuable new data, but leaves several questions still open.

We now provide information about the accuracy of testosterone and corticosterone treatment in sex ratio adjustment, and discussion about how likely it is that these hormones are skewing sex ratios. Generally, in our previous studies, the manipulations that we use here triggered 70% of hens to skew sex ratios in one direction or the other (depending on the hormone). We attempted to account for the 30% that don’t by pooling the germinal discs with the hope that the majority (70%) of the germinal discs in each pooled sample would be from hens that skewed their sex ratios in response to the treatment. Given that we now have gene pathways and genes that were differentially expressed in response to these treatments, the next step is to examine whether manipulation of those genes actually results in se ratio skews. See discussion (Lines 360-375) for our added discussion of this). 

The experimental design is appropriate and sample sizes are good. The pilot study reassures that hormonal injections are effective. Molecular analyses are carried out up the standards.

I have reviewed this manuscript before, and currently, I was happy to find that the presentation of the study has greatly improved. Few remaining issues are listed below.

Line 70 – Sex steroids and glucocorticoids can influence telomere length. Please specify the tissue in which the cited references studied telomere length.

We now specify the tissue in which telomere lengths changed (lines 71-75)

Line 101-2- The current study does not look at the differential segregation of the sex chromosomes.

Thank you for catching this. We have now reworded the sentence indicated. 

Line 225 – Please check degrees of freedom for the denominators.

We have checked and corrected the degrees of freedom for the denominators. Thanks for catching this error. 

Line 646 – Should be “selected”, this also refers also lines 653, 657.

We have made this change. 

Supplementary materials are incomplete – original data for plasma hormone levels following injection of the hens is not provided.

We have now uploaded the data for the plasma hormone injections as the supplementary materials.

Journal Requirements:

We have checked this.

2. In your Methods section, please provide additional details regarding the animals used in your study and ensure you have described the source. For more information regarding PLOS' policy on materials sharing and reporting, see https://journals.plos.org/plosone/s/materials-and-software-sharing#loc-sharing-materials."

We now include details about where we got our birds. 

3. As part of your revision, please complete and submit a copy of the ARRIVE Guidelines checklist, a document that aims to improve experimental reporting and reproducibility of animal studies for purposes of post-publication data analysis and reproducibility: https://www.nc3rs.org.uk/arrive-guidelines. Please include your completed checklist as a Supporting Information file. Note that if your paper is accepted for publication, this checklist will be published as part of your article.

We have completed and uploaded this checklist.

4. In your Methods section, please include a comment about the state of the animals following this research. Were they euthanized or housed for use in further research? If any animals were sacrificed by the authors, please include the method of euthanasia and describe any efforts that were undertaken to reduce animal suffering.

We now include a comment that birds not used for this work were transferred to another project. 

5. We note that you are reporting an analysis of a microarray, next-generation sequencing, or deep sequencing data set. PLOS requires that authors comply with field-specific standards for preparation, recording, and deposition of data in repositories appropriate to their field. Please upload these data to a stable, public repository (such as ArrayExpress, Gene Expression Omnibus (GEO), DNA Data Bank of Japan (DDBJ), NCBI GenBank, NCBI Sequence Read Archive, or EMBL Nucleotide Sequence Database (ENA)). In your revised cover letter, please provide the relevant accession numbers that may be used to access these data. For a full list of recommended repositories, see http://journals.plos.org/plosone/s/data-availability#loc-omics or http://journals.plos.org/plosone/s/data-availability#loc-sequencing.

These data have been uploaded to Gene Omnibus and we have included an accession number 

We have removed this statement. 

---

## [Decision Letter · Decision Letter 1]

8 Apr 2020

Corticosterone and testosterone treatment influence expression of gene pathways linked to meiotic segregation in preovulatory follicles of the domestic hen

PONE-D-19-33739R1

Dear Dr. Navara,

We are pleased to inform you that your manuscript has been judged scientifically suitable for publication and will be formally accepted for publication once it complies with all outstanding technical requirements.

With kind regards,

Aneta Agnieszka Koronowicz, PhD

Academic Editor

PLOS ONE

Additional Editor Comments (optional):

Reviewers' comments:

Reviewer's Responses to Questions

**Comments to the Author**

1. If the authors have adequately addressed your comments raised in a previous round of review and you feel that this manuscript is now acceptable for publication, you may indicate that here to bypass the “Comments to the Author” section, enter your conflict of interest statement in the “Confidential to Editor” section, and submit your "Accept" recommendation.

Reviewer #1: All comments have been addressed

Reviewer #2: All comments have been addressed

2. Is the manuscript technically sound, and do the data support the conclusions?

Reviewer #1: Yes

Reviewer #2: No

3. Has the statistical analysis been performed appropriately and rigorously? 

Reviewer #1: Yes

Reviewer #2: Yes

4. Have the authors made all data underlying the findings in their manuscript fully available?

Reviewer #1: Yes

Reviewer #2: Yes

5. Is the manuscript presented in an intelligible fashion and written in standard English?

Reviewer #1: Yes

Reviewer #2: Yes

6. Review Comments to the Author

Reviewer #1: I have no additional comments or remarks. My suggestions were taken into consideration by Authors.

Reviewer #2: (No Response)

7. PLOS authors have the option to publish the peer review history of their article (what does this mean?). If published, this will include your full peer review and any attached files.

Reviewer #1: No

Reviewer #2: No

---

## [Editor Report · Acceptance letter]

17 Apr 2020

PONE-D-19-33739R1 

Corticosterone and testosterone treatment influence expression of gene pathways linked to meiotic segregation in preovulatory follicles of the domestic hen 

Dear Dr. Navara:

I am pleased to inform you that your manuscript has been deemed suitable for publication in PLOS ONE. Congratulations! Your manuscript is now with our production department. 

With kind regards,

on behalf of

Prof. Aneta Agnieszka Koronowicz 

Academic Editor

PLOS ONE